# Towards a Domain-Specific Approach Enabling Tool-Supported Model-Based Systems Engineering of Complex Industrial Internet-of-Things Applications

**Christoph Binder** [1,*] **, Christian Neureiter** [1] **and Arndt Lüder** [2]

1   Josef Ressel Centre for Dependable System-of-Systems Engineering, Salzburg University of Applied Sciences, Urstein Sued 1, A-5412 Puch, Austria; christian.neureiter@fh-salzburg.ac.at
2   Institute of Ergonomics, Manufacturing Systems and Automation (IAF), Otto-v.-Guericke University, Universitätsplatz 2, D-39106 Magdeburg, Germany; arndt.lueder@ovgu.de
*   Correspondence: christoph.binder@fh-salzburg.ac.at

**Abstract:** Contemporary manufacturing systems are undergoing a major change promoted by emerging technologies such as Cyber-physical Systems (CPS) or the Internet of Things (IoT). This trend, nowadays widely known by the term "Industry 4.0", leads to a new kind of automated production. However, the rising number of dynamically interconnected elements in industrial production lines results in such a system being transformed into a complex System of Systems (SoS). Due to the increasing complexity and the challenges accompanied by this change, conventional engineering methods using generic principles reach their limits when developing this type of systems. With varying approaches only trying to find a solution for small-scaled areas of this problem statement, the need for a holistic methodology becomes more and more obvious. Having recognized this issue, one of the most promising approaches has been introduced with the Reference Architecture Model Industry 4.0 (RAMI 4.0). However, in the current point of view, this domain-specific architecture framework is missing specifications to address all aspects of such a critical infrastructure. Thus, this paper introduces a comprehensive modeling approach utilizing methods applied in Model-Based Systems Engineering (MBSE) and including domain-specific particularities as well as architectural concepts with the goal to enable mutual engineering of current and future industrial systems. The resulting artifacts, a domain-specific language (DSL), an architecture definition and a development process, are thereby consolidated in a ready to use software framework, whose applicability was evaluated by a real-world case study.

**Keywords:** Industry 4.0; systems architecture; System of Systems (SoS); Industrial Internet of Things (IIoT); Reference Architecture Model Industry 4.0 (RAMI 4.0); Model-Based Systems Engineering (MBSE)

## 1. Introduction

The need for optimized production processes with the goal to manage resources best possible force most manufacturing companies to constantly improve in order to remain competitive. The latest results from research and development offer new possibilities to support this goal, which drive change in the present industrial area and lead the path to a new form of automation-driven industry, today widely known by the term "Industry 4.0". One of the key factors ensuring the application of this trend is the emergence of Industrial Internet of Things (IIoT), an alignment of Internet of Things (IoT) to the industrial area aiming to pursue automation and data exchange in manufacturing processes [1]. The resulting interconnection of these mainly intelligent units, so-called Cyber-physical System (CPS), forms a service-oriented value creation network [2], which drifts away from the original product orientation towards technology-oriented services [3]. However, accompanied by all the opportunities for developing future industrial systems according to this trend, there are several challenges that need to be addressed in order to ensure this realization. More

precisely, as explained in [4], a major issue concerning the interconnection of components is their coexistence and interoperability, as most of them are making decentralized decisions to find the best solution for themselves. Taking this into further consideration, a new level of complexity arises when it comes to describe the architecture of current and future manufacturing systems. According to the classification scheme introduced in [5], a traditional production line can be considered a complicated system due to the large number of machines or their dynamic utilization according to what should be produced. However, by following this principle and through the integration of IIoT aspects, contemporary manufacturing systems have to be classified as complex systems, which is mainly attributed to the composition of multiple interdisciplinary elements within the production line and their possibility to make decisions on their own. In addition, falling back on the criteria mentioned in [6] as well as a CPS being a system itself, even the term System of Systems (SoS) is suggested to be used in order to emphasize the autonomous character of the system's individual components, which is substantiated by one of the early definitions of Industry 4.0 [7].

Summarizing these concerns and with regard to the arising complexity, it becomes clear that suitable ways for structuring such systems according to the different aspects addressed by Industry 4.0 need to be available. Thus, multiple organizations recently proposed different types of architectural models dealing as a reference for describing an industrial system. A collection is presented in [8], where the single projects are also described in more detail. While each reference architecture has different objectives and therefore provides different viewpoints, the main targets concerning flexibility, usability and productivity can be found in all approaches. With special focus on the mentioned aspects and due to its maturity level, Reference Architecture Model Industrie 4.0 (RAMI 4.0) [9] stands out in particular. This can be attributed to the extensive methods offered for developing concrete industrial system architectures. Those are more precisely described in two separate contributions [10,11], where the framework is also compared to another promising approach, Industrial Internet Reference Architecture (IIRA). However, although RAMI 4.0 is already used in several projects and found its way to standardization, it is still difficult to develop a specific system architecture according to its concepts at the current point of view. One of the main reasons causing this issue is the missing formalization in terms of architectural specifications on each of its layers, impeding the utilization of Model-based Systems Engineering (MBSE) concepts. More precisely, currently only a rough frame to work in is provided, resulting in a concept looking good on paper without concrete applicability.

Therefore, this paper deals with three major contributions. First, the architectural model of RAMI 4.0 is precisely refined by making use of the ISO 42010 and elaborating viewpoints to address the stakeholder concerns. To enable MBSE according to those viewpoints, the next step is to design a Domain-specific Language (DSL) based on Systems Modeling Language (SysML). Subsequently, based on the developed artifacts, the second contribution is to provide a specific systems development process, derived from the ISO 15288. The resulting process model should unite aspects originating from the architecture definition with the concepts applied by methodologies used in MBSE, similar to Model-driven Architecture (MDA). The last contribution is to evaluate the developed artifacts and validate the feasibility of the approach by applying a real-world case study, a manufacturer of subway tracks.

To address these aspects, the remainder of this contribution is structured as follows. Section 2 provides an overview of RAMI 4.0, used standards and technologies as well as current approaches applied in the engineering of IIoT-based systems. In Section 3, the approach itself to challenge the mentioned problem is stated. Next, the development of the previously mentioned artifacts is explained in detail in Section 4, whose applicability is demonstrated with a typical industrial use case in Section 5. Finally, in Section 6, the results of the conducted study are summarized and a conclusion is given.

## 2. Related Work

In this section, the theoretical background of the intended approach as well as its related work is explained in more detail. To do so, first an overview of state-of-the-art reference architectures is given within the first subsection. Subsequently, contemporary approaches and methodologies in the area of model-based industrial systems engineering are listed and further analyzed.

### 2.1. Domain-Specific Architecture Frameworks

The goal of RAMI 4.0 is to enable the discussion of an Industry 4.0 system based on domain-specific viewpoints. The three-dimensional model, delineated in Figure 1, has been mainly developed to create a common understanding and a mutual basis for industrial systems engineering [9]. Due to the big influence of its creators on the German industry, the reference architecture encloses multiple sectors within the industrial area and even has been standardized in the standard DIN SPEC 91345 [12]. Moreover, a system either is developed as a whole or single parts of it are considered in more detail, according to the corresponding Industry 4.0 related use case. In more detail, the horizontal axis of RAMI 4.0, the so-called "Life Cycle and Value Stream", deals with the different states an asset may have during its time of usage by falling back to the criteria introduced in the standard IEC 62890. In the second axis, the vertical integration within a factory is represented by the "Hierarchy Levels" based on IEC 62264 and IEC 61512, better known by the term "Automation Pyramid". Finally, the top-down arrangement of the layers enables the structuring of the system according to the feature of its components across six *Interoperability Layers*.

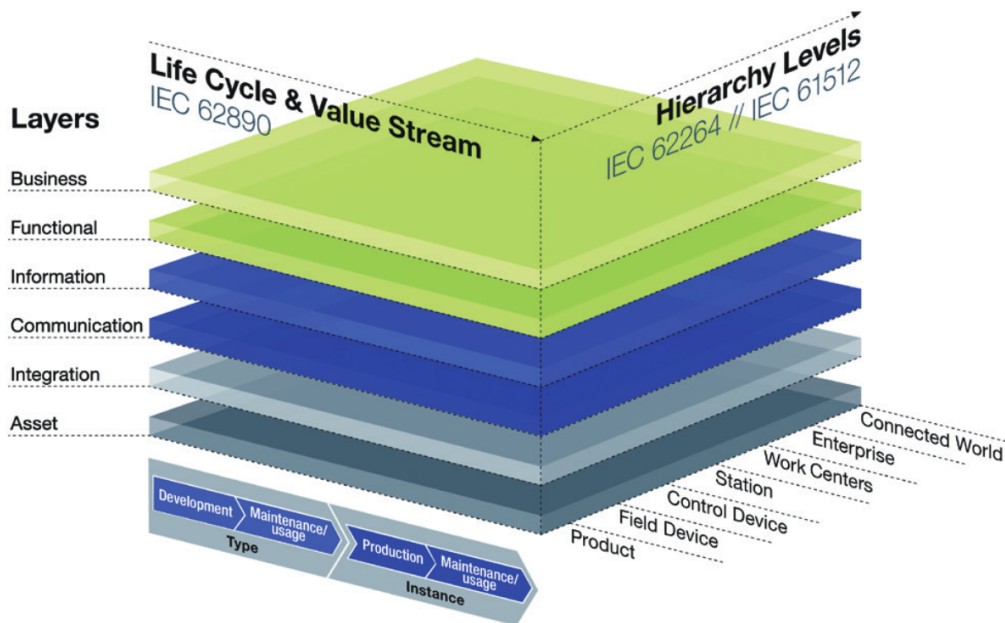

**Figure 1.** Reference Architecture Model Industry 4.0 (RAMI 4.0) [13].

Parallel to the German approach, a similar concept was proposed by the Industrial Internet Consortium (IIC) with the name of IIRA [11]. In comparison to RAMI 4.0, this reference architecture also focuses on the different interests of stakeholders within an industrial system. By doing so, it makes use of the ISO 42010 as a guideline for building an architecture in a particular domain or a community of stakeholders. Thus, these conventions and principles have a significant influence on an industrial model based on IIRA. However, in contrast to RAMI 4.0, only four levels to reflect the stakeholder concerns are defined within the architectural model. The Business and the Functional Viewpoint can thereby be compared to the equally called RAMI Layers. In contrast, the Usage as well as

the Implementation Viewpoint deal with describing the system during its run-time and its implementation, as the names imply.

Additionally, the European joint project has set itself the task of ensuring collaborative automation using different kinds of networks and embedded devices in five domains, resulting in the proposal of the so-called Arrowhead Framework. The domains include home and industrial automation, production, virtual markets of energy and electrical vehicles infrastructures [14]. The primary goal of this approach is to address the technical difficulties that come with collaboration issues in automation. Thus, a framework for the modeling of systems has been adopted to this objective, which is provided in order to advance innovations and standardization. In [15], the service-oriented architecture of the framework and its core components are introduced. The main contribution of this work is the possibility to enable the interoperability between systems originating from the previously mentioned domains by considering them from a SoS perspective.

### 2.2. Model-Based Systems Engineering

In summary, the approaches above build a promising foundation for structuring and enabling the discussion of future industrial systems. However, as all of the mentioned reference architectures were proposed in recent years, they are more or less theoretical concepts, which look good on paper, but are lacking concrete applications in order to guide the development of an Industry 4.0 based system. For example, RAMI 4.0 mainly specifies the frame to work in giving a rough outline for the boundaries of the system, but does not define a specific development process. Thus, different projects trying to actually implement applications according to this reference architecture have emerged. The first steps to understanding and realizing the vision of digitalizing production and manufacturing in those systems is proposed in [16] by modeling the digital twin of a CPS in regard to concrete standards and technologies such as those introduced by the administration shell of RAMI 4.0 [17]. However, as explained above, MBSE is one of the chosen methods concerning consistency within a model of a constantly changing system by keeping traceability at the same time. Thus, several projects try to provide MBSE methods such as modeling languages for addressing different aspects when developing future industrial systems, based on Unified Modeling Language (UML), SysML or even defining a DSL [18–20].

Similar to the projects tailored to RAMI 4.0, model-based systems engineering approaches for developing Industry 4.0 based systems have been proposed for other reference architectures. Thus, an approach to model and develop a CPS-based manufacturing system from a technical perspective is proposed in [21]. To mention another example, the authors of [22] described an approach for the development of IIoT applications by considering them as a SoS and making use of the concepts of IIRA. A special feature of their work is the mapping of the IIRA viewpoints to those of the Unified Architecture Framework (UAF), which enables MBSE by applying the extensive possibilities for model-based systems development of this framework. Furthermore, Radanliev [23] used cyber-elements for Industry 4.0, collected from different trends throughout global regions. The goal of this approach is to summarize those elements to a framework for describing IoT-based cyber-security models as part of a more comprehensive reference architecture [24]. As far as the Arrowhead Framework is concerned, a project has been proposed making this framework the basis for realizing the supply chain of a paper-making company [25]. The main goal of this work is to identify impacts of Industry 4.0 on Supply Chain Management (SCM) as well as logistics in order to support these business units in digitalization issues . Additionally, several projects introduce applications supporting the engineering process within industrial systems by applying decision trees or security frameworks [26–28].

However, all of the mentioned approaches are more or less missing applicability aspects, as no user is able to develop the respective system with a suitable tool. Thus, to fully enable the applicability of MBSE, a comprehensive framework inheriting a specific and detailed development process, integrating well-known and tool-supported methods, needs

to be provided. In conclusion, the need for a methodology tailored to future industrial systems engineering becomes more and more obvious.

## 3. Approach

As explained in above, the main purpose of this work is to provide the possibility for developing industrial systems based on RAMI 4.0 in order to ensure the applicability of the proposed theoretical reference architecture. Nevertheless, actual applications are a rarity in the current point of view, although RAMI 4.0 is one of the most promising approaches when it comes to handling the complexity of Industry 4.0-based systems. Thus, a software tool is developed, which addresses aspects of current characteristics in the area of systems engineering by considering peculiarities of the industrial domain at the same time. As both MBSE and Industry 4.0 are growing areas, which are dynamically altering or profiting from new advances in research and development, suitable methods for developing the piece of software need to be applied. Hence, the Agile Design Science Research Methodology (ADSRM) appears to be the right method to be utilized in such dynamic application scenarios. Thereby, this agile methodology introduces five process steps, as visualized in Figure 2. By offering multiple entry steps into its iteration cycle, the whole process is supported by exploratory case studies [29]. In this specific work, the development cycle is initiated by defining a case study. Based on the chosen use case, requirements can be derived for specifying quantitative information about what should be developed and how to approach this. Based on those requirements, so-called artifacts can be developed. In this case, an architecture definition, a process model as well as method integration are considered to be such artifacts. Finally, the case study is implemented in order to evaluate and verify the developed components, whose result will then serve as basis for the next iteration step.

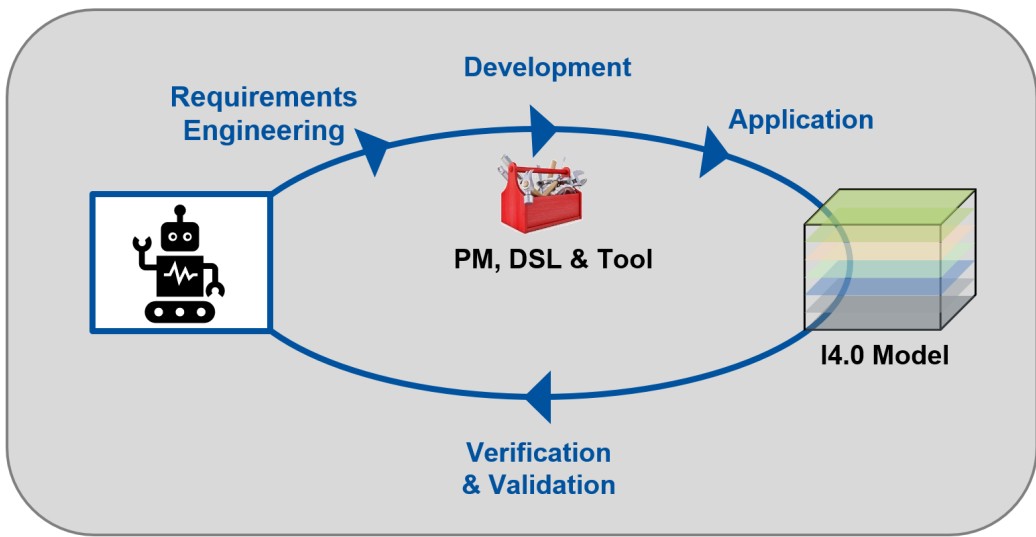

**Figure 2.** Agile Design Science Research Methodology (ADSRM).

*Case Study Design*

The proposed work makes use of a typical industrial case study, a manufacturer of subway tracks. Specific information such as business models or production line infrastructures is thereby provided by a company partner, combined with the desired need for a transformation inheriting the concepts of the fourth industrial revolution. Thus, to integrate Industry 4.0-related aspects, the single subway tracks are produced in sample size 1. In addition, the whole life-cycle of the subway track has to be considered as well as state-based maintenance or any supply-chain challenges. As the system of this case study should be developed based on the specifications of RAMI 4.0, the requirements need to be derived in the next step in order to fulfill the ADSRM specification as well as provide

fundamental directions for creating the piece of software. In this particular scenario, the intention of modeling the case study should consider the following requirements:

- Integrate and evaluate the ISO 42010 for refining the architecture of RAMI 4.0.
- Follow systems engineering according to a particular development process.
- Validate the applicability of the RAMI Toolbox.
- Execute a feasibility analysis for future and more sophisticated projects.
- Develop functionalities for automating repetitive tasks.

## 4. Implementation

In this section, the implemented steps for developing the intended modeling software, better known by the term "RAMI Toolbox", are explained in more detail. As already mentioned, this task is constituted of three different parts. First, a comprehensive architectural definition of RAMI 4.0 is given by also defining a DSL and using established standards. Subsequently, a specific development process is defined and useful software functions are implemented into the RAMI Toolbox. As this piece of software is aims at being an extension for the modeling software Enterprise Architect (EA), all following concepts are tailored to work in this environment and therefore inherit tool-specific conventions.

### 4.1. Architecture Development

The first step to create the RAMI Toolbox is to define the RAMI 4.0 architecture based on the ISO 42010. This will result in having a standardized backbone for the following domain-specific adaptations. To look further into the theoretical concept of the standard, there are multiple stakeholders having interest in the architecture of a system, described as concerns. The goal of the architecture is to define viewpoints in order to address those stakeholder concerns. However, as the specification of RAMI 4.0 provides several interoperability layers, those can directly be converted into viewpoints. Nevertheless, this is the point where the reference architecture is lacking information. To counteract this issue, the next step of the ISO 42010 aims to define model kinds in order to provide applicable instruments for practically implementing the viewpoints. Thus, with regard to the concerns mentioned in the official RAMI 4.0 specification document and the corresponding viewpoints [9], the views and model kinds delineated in Table 1 have been specified.

**Table 1.** RAMI 4.0 viewpoints and model kinds.

| Viewpoint | View | Model Kind |
|---|---|---|
| Business Layer | Context View | SIPOC Model |
| | Process View | BPMN Model |
| | | Value Chain Model |
| | Business View | Business Case Model |
| | | Goal Model |
| | Requirements View | Requirements Model |
| Function Layer | FAS View | Use Case Model |
| | | Functional Development |
| | Function View | Black Box Model |
| | | White Box Model |
| | Logical View | Actor Mapping Model |
| Information Layer | Information View | Information Model |
| | Data View | Data Model |
| Communication Layer | Communication View | Communication Model |
| | Interface View | Interface Model |
| Integration Layer | Technical View | Component Model |
| | | ICT Model |
| | | HMI Model |
| Asset Layer | Realization View | Asset Model |
| | | RTE Model |

Based on the elaborated model kinds, the architectural model of an industrial system based on RAMI 4.0 can be created with the help of the RAMI Toolbox. To do so, the last piece is missing, the specification of a particular DSL. By deriving elements from the UML or SysML, the needed semantics and structure for understanding the application domain as well as the physical world of Industry 4.0 is defined. As the development of the DSL is precisely defined in [30] and its usage is explained in detail in Section 5, it is not outlined any further at this point.

### 4.2. Process Model Definition

As there are many aspects to consider throughout the whole engineering life-cycle of RAMI 4.0-based systems, a clear development process guiding users executing this task has to be provided. Therefore, this work also introduces a specific process model utilizing the well-known engineering standard ISO 15288. Theoretically, as Lake explained, the development of such a complex system needs to be structured according to the phases of its life-cycle in order to prevent confusion, misunderstanding or even conflict [31]. This is why the authors of [32] originally proposed a process model for developing Industry 4.0 applications based on RAMI 4.0, which is extended according to novel advances in this work. To give a small excerpt, Figure 3 provides an overview of how the process model correlates with the architecture definition.

At the top of the image, the two processes *Stakeholder Needs Definition Process* and *Requirements Analysis Process*, which are defined in the ISO 15288, are combined to the RAMI 4.0 specific *Business Analysis Process*. As recognizable from its name, this process deals with developing the Business Layer of the architecture. Subsequently, single process steps are elaborated, which indicate executable tasks for transforming all inputs from the System of Interest (SoI) to its output. By doing so, every single step is performed in the context of a view and results in creating one or more architectural models.

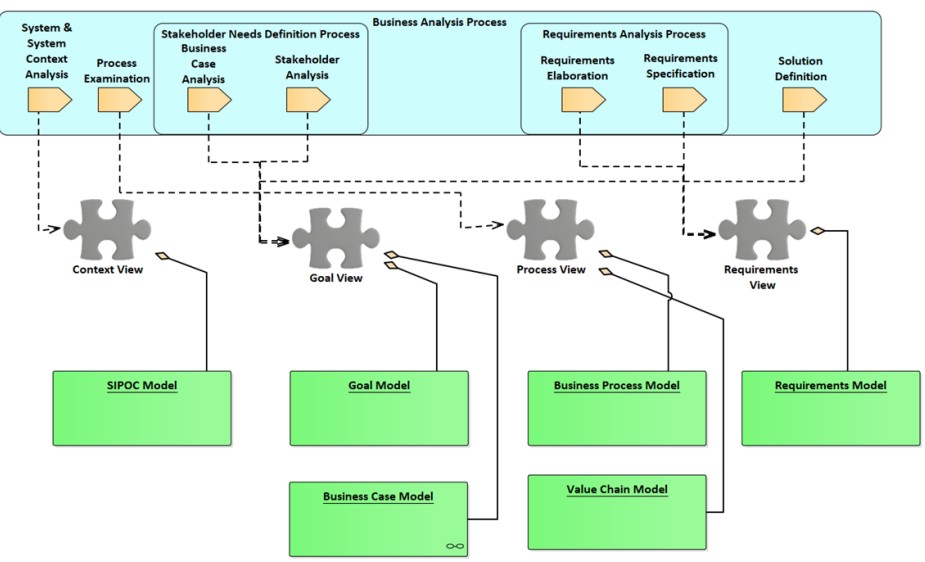

**Figure 3.** Process model for developing RAMI 4.0 Business Layer.

*4.3. Framework Integration*

The Functional Architecture for Systems (FAS) method was introduced by Weilkiens [33] due to lack of common approaches for functional architecture development, in particular in the context of MBSE. However, especially in the industrial domain, systems strongly rely on their functions, since they ensure the traceability between the requirements and the actual implementation. In more detail, in complex systems, a single function usually is deployed on a large number of technical components, while a single component is able to carry out more than one function. With the FAS method, a methodology for developing the architecture of such interwoven systems has been introduced. According to the aforementioned reasons, the eligibility for this method to be applied in the context of RAMI 4.0 becomes clear and is therefore utilized to further refine the Function Layer.

Additionally, it has been pointed out that the cubic layout of RAMI 4.0 is missing one abstraction level. In particular, when describing a whole production system, the combination of both reference architectures reaches its limits. On the other hand, single CPS could be developed in detail throughout the whole life-cycle including aspects such as Round-trip Engineering (RTE) or application in a co-simulation environment. Thus, in the current version of the toolbox, the Software Platform Embedded Systems (SPES) method is implemented to counteract with those issues. However, in future versions, a more comprehensive methodology has to be integrated or developed by itself.

In addition, a detailed description how the mentioned concepts are used within the RAMI Toolbox is given by the case study illustration in Section 5.

## 5. Application

In this section, the development of the case study (a click-through model is available at http://www.rami-toolbox.org/UseCaseATS, accessed on 23 March 2021) is described in more detail. The main goal of this case study is to evaluate the created process model, the architecture definition as well as the DSL and verifying its potential to create industrial sys-

tems based on RAMI 4.0. To do so, the use case describing the development of individual subway tracks is delineated in detail according to the previously mentioned aspects.

This means, the first step is to describe the Business Layer of RAMI 4.0. In the first task, the system context of the SoI has to be defined, which is done with the help of a SIPOC-Diagram (suppliers, inputs, process, output and customers). The SoI is considered to be a process transforming the inputs to the outputs. In the illustrated case study, the SoI is represented by the production process of the subway track itself, as outlined in detail in Figure 4. The diagram indicates this model in a simplified way on the highest perspective by summarizing all material suppliers as well as all energy providers. An example for a further input are considered to be the product requirements from the product engineering department, while the final result is transmitted to the laboratory as output of the SoI.

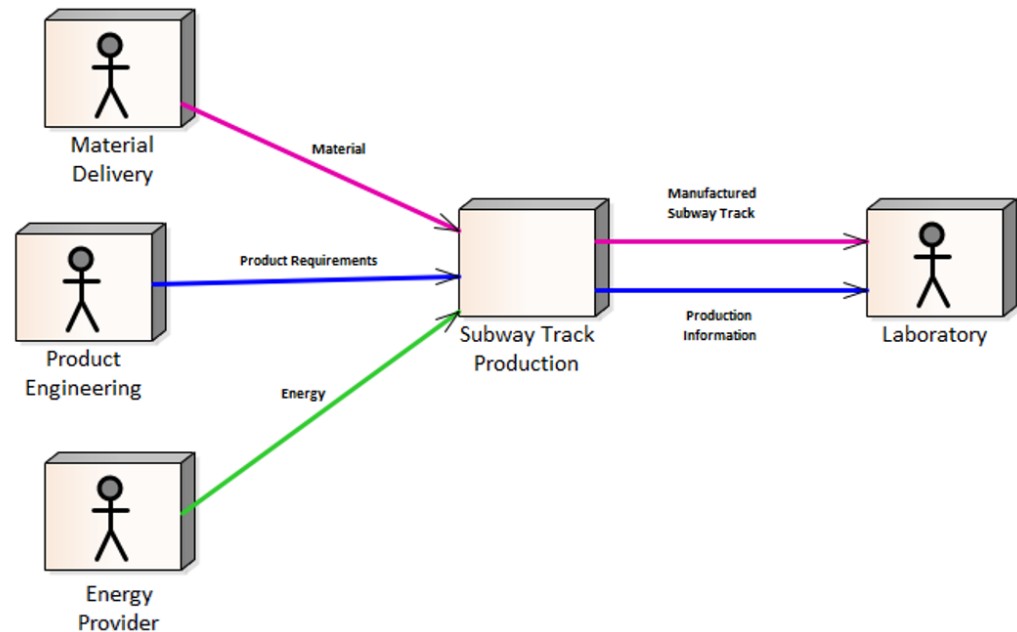

**Figure 4.** RAMI 4.0 SIPOC model.

Thus, the next step is to indicate how this process is executed, in this case with the help of a Business Process Model and Notation (BPMN) diagram. On different abstraction levels, all business or manufacturing processes being connected with the SoI in any possible way are depicted. A specialty of the RAMI Toolbox is the provision of a value chain model for modeling the manufacturing processes, as depicted in Figure 5. In this model, material and information flows are shown in the so-called value-stream mapping, which is especially targeted to accompany the development of manufactured goods. In this case, the production of the metal profile for the subway track, which is demonstrated by the bottom sequence of events in the image, is outlined in the image. On top, the information flow is shown by the blue arrows, while the red arrow represents a manual stream. In addition, the two lightnings stand for the so-called *Kaizen Bursts*, which are used to indicate optimization potential or susceptibility to errors. In this case, the information flow on top of the image can be automated and the mechanical flow is prone to errors.

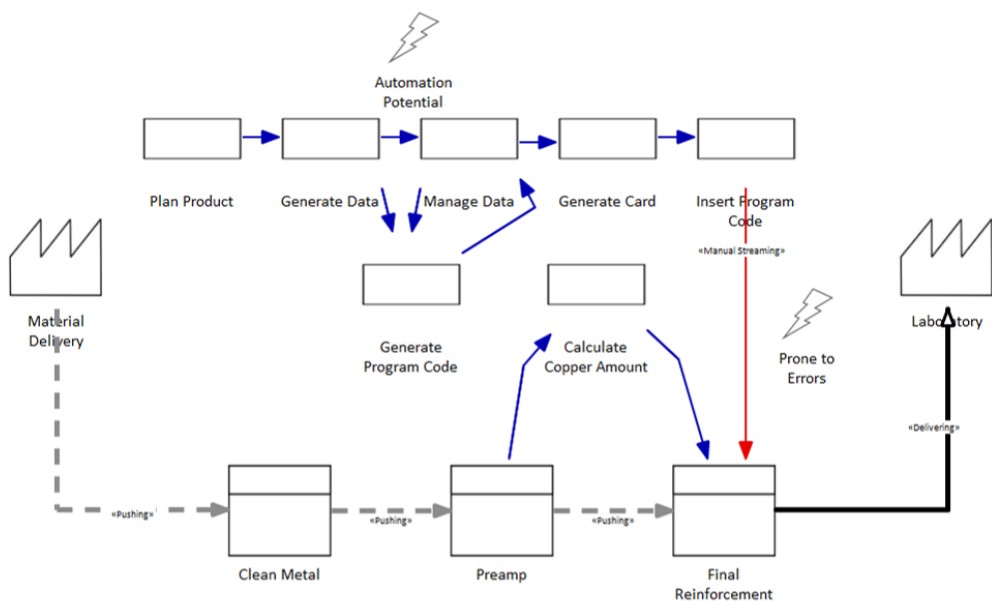

**Figure 5.** RAMI 4.0 value-stream mapping.

Based on this analysis, new business cases are derived from the single Kaizen Bursts, which are used to describe ways for integrating Industry 4.0 aspects to optimize the production process. Those business cases exhibit an interconnection to stakeholders having interest in the respective SoI. With regard to all interests coming from the single business actors, a comprehensive requirements analysis can be performed, serving as basis for further system developments.

As previously mentioned, the interface between the requirements and the physical components of the system is realized with the FAS method, by defining functions fulfilling the requirements and being deployed to system parts. The first step to achieve this is to model all desired processes within the business case with the help of activity diagrams. All atomic process steps, called actions, are thereby summarized and represent the tasks to be executed by the system. Similar tasks are then summarized to functional groups, which are realized by functional elements. Those elements represent the functions of the system, as depicted in Figure 6. By showing them in a black box or white box perspective, their interconnection and exchanged information can be indicated as well as interference or disturbances influencing the transformation from the input to the output. The displayed image however reveals the interfaces between the single functions and their data exchange, such as the length of the modules for calculating the content of copper needed for the electrical conductors. The last step of the Function Layer is to deploy those functions to logical elements representing the physical system parts. This finally ensures the traceability between the requirements and the actual system components, since it is possible to fall back which requirement is fulfilled by which component.

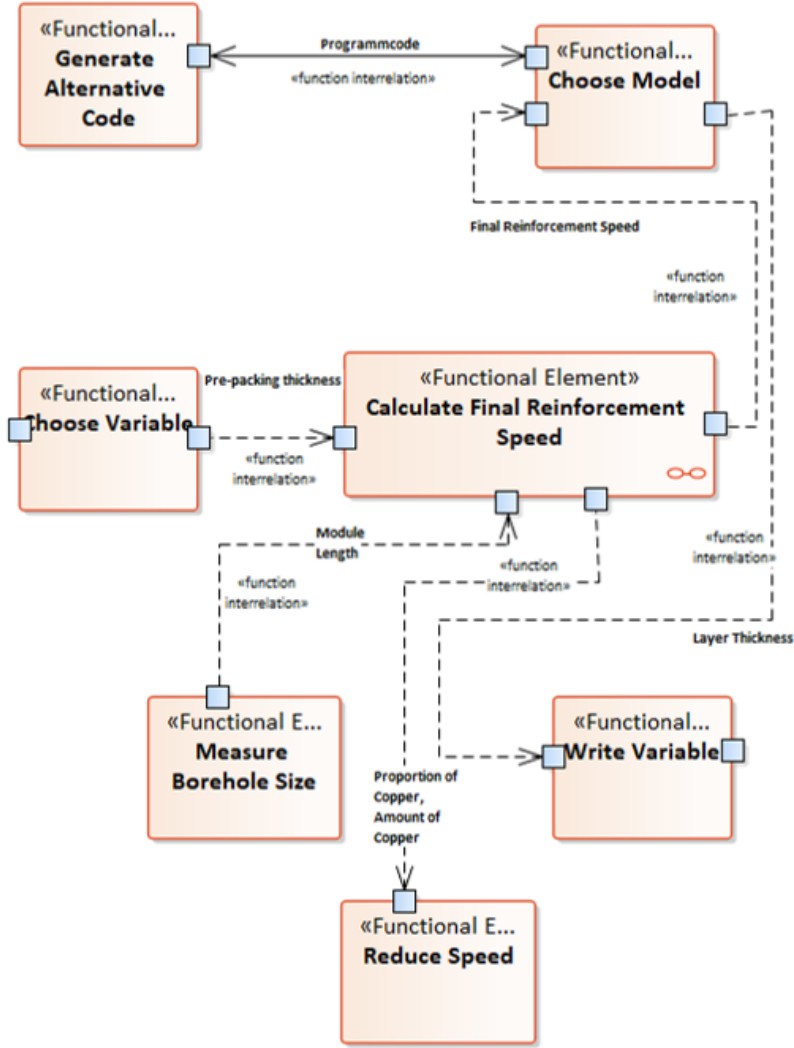

**Figure 6.** RAMI 4.0 Function Layer.

From this point on, the Information and the Communication Layer can be described in more detail, by modeling what is exchanged between the components and how it is done. Therefore, a data flow diagram is used to describe the exact information flow with all information items, data stores and external data sources. The next big step is to define data model standards such as JSON or XML to indicate the format of the single information that is exchanged. Based on this information, the interfaces in the Communication Layer can be adapted. As the information object remains the same, the direction flow is now specified using request or service points. Additionally, the protocols ensure that the data model standard is correctly transmitted and furthermore define the interfaces between the single components. In Figure 7, a diagram considering the just mentioned aspects is delineated. It indicates that the measurement data are transmitted via Near-field Communication (NFC), while machine codes are sent with the help of Ethernet in this particular case study. This falls back to the fact that the current infrastructure only allows this type of protocol for information exchange. In future iterations of ADSRM, this could be investigated further and novel technologies or protocols such as Open Platform Communications Unified Architecture (OPC UA) might be implemented. The main control unit however needs to have multiple interfaces, including non-technical ones such as a Human-machine Interface (HMI). Additionally, in the Communication Layer, all networking information is further defined. While one particular granularity level, as shown in the mentioned image, depicts the used system components and their interconnection with interfaces, other levels

deal with the detailed specification of those interfaces. For example, with regard to the Ethernet interface, a higher level considers the type of communication network standard, such as PROFINET or Ethernet for Control Automation Technology (EtherCAT), as used within the complete production system. On a lower granularity level, more details about the single protocol transmitting the data between the industrial devices are stated. While the data are depicted as Information Objects, protocols such as NFC or 4G indicate how these technologies are used to send the objects over the interfaces within the company network. Subsequently, a lower level allows a more detailed insight within a single protocol and could enable model-based implementations or code generation.

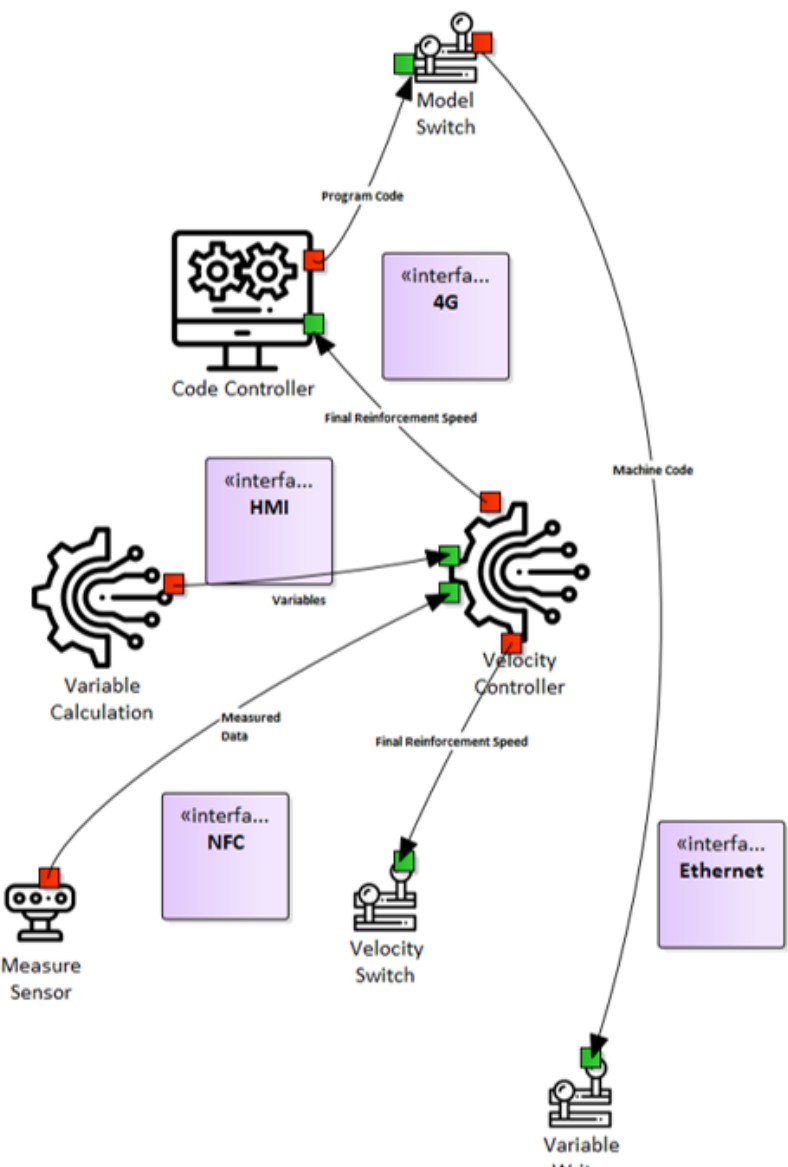

**Figure 7.** RAMI 4.0 Communication Layer.

The transformation from the logical perspective towards a more technical one takes place in the Integration Layer of RAMI 4.0. In this viewpoint, the administration shell of the components is modeled in detail. According to the official RAMI 4.0 documents, this assures that all data are collected throughout the whole life-cycle of an asset and even components not able to actively communicate find their place in an Industry 4.0 system. Furthermore the Integration Layer deals with specifying all HMIs as well as the Information and Communication Technology (ICT) network infrastructure. For example, the modeling

of a Supervisory Control and Data Acquisition (SCADA) system takes place at this layer. By considering information resulting from the Information as well as Communication Layer, data management as well as communication structures can be used for engineering such a SCADA system. Thus, as the vertical integration within a factory is also considered by RAMI 4.0, the work centers or station panes provide an ideal location for depicting the interfaces between organization and shop floor. On a lower granularity level, the architecture of the SCADA system itself is modeled in order to optimize its implementation. At last, the Asset Layer represents the single system components in detail considering all technical aspects as well as software specifications, which would bridge the model and the actual implementation. In addition, this viewpoint also enables RTE or the embedding of components into simulation scenarios.

*Findings*

A big advantage of the RAMI Toolbox is that the software is publicly available and open-source. This means, it might be used by multiple system engineers in various projects. The results thereby can be analyzed and lead to constantly evolving the tool towards new findings from research and industry. The iterative development process ADSRM at the same time acts as major technology driver supporting this step. New developments and research outcomes based on the RAMI Toolbox can thus be proposed to the community within short cycles. Hence, developing the case study concerning the subway tracks according to ADSRM led to some interesting results. Although the RAMI Toolbox is aiming to be a universal tool for developing any kind of industrial systems, the current specifications target the description of production lines on a higher abstraction level. As the SPES method however enables engineering of embedded systems on various hierarchy levels, due to missing specifications and inequalities or deviations with RAMI 4.0, the implementation of a proprietary method especially focused on the industrial area appears to be a preferable solution. More precisely, as an industrial system aligned to RAMI 4.0 is a SoS consisting of multiple CPS itself, each of the single subsystems has to be engineered in detail as well. Thus, by implementing a methodology extending the RAMI 4.0 cube and aiming to address the aforementioned issues, engineering of current and industrial systems could be vastly improved.

Additionally, as can be recognized from the model, the Business as well as the Function Layer of RAMI 4.0 are clearly defined with specific process steps in order to be developed. This results mainly from integrating domain-specific knowledge as well as business expertise from the various stakeholders. However, when it comes to model the system with respect to a technical perspective, the RAMI Toolbox is offering a more interpretative working environment. Additionally, this leads back to missing specifications in the RAMI 4.0 cube as well as the just mentioned problem of engineering systems on lower abstraction levels. To handle this, future research further helps refining the toolbox. On the one hand, the integration of the OPC UA leads to specifying IIoT-based communication infrastructures or data standard models, as explained is several publications [34,35]. On the other hand, AutomationML is the leading standard when it comes to storing or exchanging plant engineering information [36]. Consequently, the goal of the next iteration must be to extend the RAMI Toolbox in order to consider and utilize these technologies.

## 6. Conclusions and Future Work

Engineering of systems rising in complexity becomes an increasingly difficult task. In particular, the industrial domain is profiting from the ongoing transformation originating from the introduction of CPS, IIoT or new communication technologies. Nevertheless, as MBSE proves to be a promising way to handle the complexity of such dynamically changing systems in theory, the need for practical applications becomes obvious. For these reasons, this paper proposes an approach for developing complex IIoT applications by providing a specific toolset, the so-called RAMI Toolbox. This is done by implementing a number of needed steps, as delineated in Section 4. First, the specifications of RAMI 4.0 are

analyzed in detail and extended by the concepts of the ISO 42010. Then, a particular DSL is developed in order to allow all included stakeholders to find a mutual communication basis. To handle the complexity during the engineering task, this work also introduces a development process giving step-by-step guidelines. Finally, the developed framework is evaluated and validated towards its feasibility in Section 5. Based on the result of this work, industrial systems engineering could be taken to the next step by increasing the acceptance of RAMI 4.0 in the community and as a consequence its utilization in industrial projects.

However, this approach must not be seen as a ready-to-use methodology but rather a next step in the right direction. Several follow-up projects could further refine and enhance the RAMI Toolbox. To mention some examples, to enable development of embedded units, further specifications have to be integrated into the software. Furthermore, on the bottom layers of RAMI 4.0, additional optimizations need to be performed. At the current point of view, the model is more or less an abstract illustration of an industrial system, without the possibility to go too much into detail. The integration of accepted standards such as OPC UA or AutomationML could help in this matter, which would highly contribute to physical asset development or system evaluation. Overall, the complete and refined approach needs to be verified and evaluated with more sophisticated case studies in order to guarantee its seamless application.

**Author Contributions:** C.B. declared the problem definition, conducted the study, discussed the results and contributed in writing and preparing the original draft of the manuscript. Thereby, C.B. furthermore dealt with the conceptualization of the study, the definition of the methodology, the development of the software and investigating as well as validating the resulting artifacts. C.N. and A.L. provided the initial formal problem definition and contributed academically by offering support and preliminary works. All authors have read and agreed to the published version of the manuscript.

**Funding:** This research was funded by the "Christian Doppler Forschungsgesellschaft".

**Data Availability Statement:** The architectural model was created with the help of Enterprise Architect and is publicly available at https://rami-toolbox.org/UseCaseATS/ (accessed on 23 March 2021). The code developed as a result of the present contribution is implemented as part of the RAMI Toolbox, which is freely available at https://rami-toolbox.org/download/ (accessed on 23 March 2021). Additionally, all data generated and analyzed in this contribution are available from the corresponding author on reasonable request.

**Acknowledgments:** The support for valuable contributions of LieberLieber Software GmbH and successfactory consulting group is gratefully acknowledged. The financial support by the Austrian Federal Ministry for Digital and Economic Affairs, the National Foundation for Research, Technology and Development and the Christian Doppler Research Association and the Federal State of Salzburg is also gratefully acknowledged.

**Conflicts of Interest:** The authors declare no conflict of interest.

## Abbreviations

The following abbreviations are used in this manuscript:

| | |
|---|---|
| RAMI 4.0 | Reference Architecture Model Industrie 4.0 |
| IoT | Internet of Things |
| IIC | Industrial Internet Consortium |
| IIoT | Industrial Internet of Things |
| SoS | System of Systems |
| ADSRM | Agile Design Science Research Methodology |
| MBSE | Model-based Systems Engineering |
| MDA | Model-driven Architecture |
| DSL | Domain-specific Language |
| DSSE | Domain Specific Systems Engineering |
| UML | Unified Modeling Language |
| CPS | Cyber-physical System |
| FAS | Functional Architecture for Systems |

| | |
|---|---|
| HMI | Human–machine Interface |
| EA | Enterprise Architect |
| SoI | System of Interest |
| SPES | Software Platform Embedded Systems |
| IIRA | Industrial Internet Reference Architecture |
| UAF | Unified Architecture Framework |
| SysML | Systems Modeling Language |
| ICT | Information and Communication Technology |
| SCM | Supply Chain Management |
| RTE | Round-trip Engineering |
| NFC | Near-field Communication |
| BPMN | Business Process Model and Notation |
| EtherCAT | Ethernet for Control Automation Technology |
| SCADA | Supervisory Control and Data Acquisition |
| OPC UA | Open Platform Communications Unified Architecture |

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
