# Peer review of "Towards a Domain-Specific Approach Enabling Tool-Supported Model-Based Systems Engineering of Complex Industrial Internet-of-Things Applications"

_systems, doi:10.3390/systems9020021_

Round 1

Reviewer 1 Report

  • The overall paper should be carefully revised with a focus on the language: especially grammar and punctuation.
  • The format of references should be identical. -Overall, there are still some major parts that the authors did not explain clearly.
  • I can see some paragraphs in the introduction, related work, and proposed approach which should be merged.
  • The objectives of this paper need to be polished.
  • figure 2 and 4 should be polished
  • The author should polish the future work and the importance of this work. (short and concise)
  • some references can be cited to border the scope of the paper

J. Wang, W. Chen, L. Wang, Y. Ren and R. S. Sherratt, “Blockchain-based data storage mechanism for industrial internet of things,” Intelligent Automation & Soft Computing, vol. 26, no.5, pp. 1157–1172, 2020.

L. Aguilar, S. W. Nava-Díaz and G. Chavira, “Implementation of decision trees as an alternative for the support in the decisionmaking within an intelligent system in order to automatize the regulation of the VOCS in non-industrial inside environments,” Computer Systems Science and Engineering, vol. 34, no.5, pp. 297–303, 2019.

W. Xu, Y. Tao, C. Yang and H. Chen, “Msicst: multiple-scenario industrial control system testbed for security research,” Computers, Materials & Continua, vol. 60, no. 2, pp. 691–705, 2019.

L. Xu, C. Xu, Z. Liu, Y. Wang and J. Wang, “Enabling comparable search over encrypted data for iot with privacy-preserving,” Computers, Materials & Continua, vol. 60, no. 2, pp. 675–690, 2019.

A. Badshah, A. Ghani, M.A. Qureshi and S. Shamshirband “Smart security framework for educational institutions using internet of things (iot),” Computers, Materials & Continua, vol. 61, no. 1, pp. 81–101, 2019.

C. Li, G. Xu, Y. Chen, H. Ahmad and J. Li, “A new anti-quantum proxy blind signature for blockchain-enabled internet of things,” Computers, Materials & Continua, vol. 61, no. 2, pp. 711–726, 2019.

Author Response

Thank you for the valuable comments on our paper. Please find attached the respective responses how your comments have been addressed in the revised manuscript.

Reviewer 2 Report

The domain is interesting. Even if the paper is written in a low granularity, which is not a very good approach for future readers, it has potential.

There are 2 issues to be approached and solved:

Issue1
The references have to be improved. It is absolutely necessary to renew and review the references.  

I consider that [1] is a very general referencing. The book is not offering a detailed or practical approach.
References [2], [5] are very old and outdated.
Regarding the rapid evolution of the domain, some references are outdated. 
Also, regarding OPC UA, reference [29] is completely outdated. There are researches right now that are covering even OPC UA Publish-Subscribe specifications.
Also, reference [30] regarding AutomationML is very old.

Issue 2
The communication has to be linked better to protocols. Communication regarding physical support is very easy thing, but communication regarding interfacing protocol is something interesting and actual in the domain. Therefore, NFC, 4G, etc., are alright to be mentioned, but the value raises in the protocols linked to the concept. 
The authors are mentioning correctly OPC UA. Yes, OPC UA is able to cover the information modelling, state based communication, etc., and the progress towards this Industry 4.0 protocol is increasing exponentially. OPC UA real-time, OPC UA and AutomationML integration, etc. are studied in the literature, and many other issues regarding 62541. Therefore, in order to increase the value of the paper, a deeper connection towards protocols, particularly OPC UA, would be necessary.  

Author Response

(The authors gave the same response as above.)

Reviewer 3 Report

The topic of the manuscript is interesting given the attention that Industry 4.0 and the RAMI 4.0 are receiving from different scientific fields, in addition, this topic fits the scope of the Journal. After a careful revision, the following comments are provided for the enhancement of the manuscript.

  • Regarding the format of the document, some suggestions are as follows.

Figure captions and titles of tables lack the terminal period (punctuation)

The acronym OPC UA is directly used without having been defined. Even, it does not appear in the abbreviations list.

References must be slightly revised to fit the format of the template. For instance, abbreviated name of journals must be used.

-           The paper is well written and organized; about the content of the manuscript, these issues are commented.

“Industry 4.0”, or “Industrie 4.0”, could be added as keyword, if the authors agree.

The contextualization is well scheduled but some relevant works have been left unnoticed. For instance, the following ones could be considered by the authors:

  • Colombo, A.W.; Karnouskos, S.; Kaynak, O.; Shi, Y.; Yin, S. Industrial Cyberphysical Systems. A Backbone of the Fourth Industrial Revolution. IEEE Ind. Electron. Mag. 2017, 11, 6–16.
  • González, I.; Calderón, A.J.; Figueiredo, J.; Sousa, J.M.C. A Literature Survey on Open Platform Communications (OPC) Applied to Advanced Industrial Environments. Electronics 2019, 8, 510.
  • Iglesias, A.; Sagardui, G.; Arellano, C. Industrial Cyber-Physical System Evolution Detection and Alert Generation. Appl. Sci. 2019, 9, 1586.
  • Tran, N.-H.; Park, H.-S.; Nguyen, Q.-V.; Hoang, T.-D. Development of a Smart Cyber-Physical Manufacturing System in the Industry 4.0 Context. Appl. Sci. 2019, 9, 3325.

A brief sentence commenting the rest of the section would facilitate the reading in section 2.

In the fifth section, page 11, it is said that “… while machine codes are sent with the help of Ethernet”. A question arises in this regard; do the authors refer to industrial communication protocols based on Ethernet? For example, PROFINET, Modbus TCP, EtherCAT, just to name a few, use Ethernet for the two lower layers of the OSI model. Some more information should be given in this sense for a proper description of the approach.

An addition question about the previous comment, should these industrial communication interfaces be considered within the Communication Layer?

HMIs are included in the presented approach, namely within the Integration Layer. In this regard, a similar function is developed by supervisory systems, commonly known as Supervisory Control and Data Acquisition (SCADA) systems, which provide visualization and also command capabilities. These systems play a paramount role in any industrial automated process, so the authors could add some brief comment about where they should be located within the RAMI 4.0.

A brief mention to the OPC UA specification is provided at the end of the fifth section, without having been previously commented. It would be interesting if this protocol was slightly commented with some more detail given the fact that it has been pointed as the main tool to handle interoperability in the Industry 4.0 area.

The paper reports a proprietary method, as pointed in the subsection 5.1. Nevertheless, as the authors know, open-source technologies are receiving important research efforts and are signaled as enabling factors for the Industry 4.0 deployment. In this sense, it would be interesting that the authors include some brief consideration about open-source systems.

Literature related to Industry 4.0 and other aligned paradigms like Industrial IoT or Smart Manufacturing usually lacks experimental implementations in order to validate the theoretical or conceptual proposals. In this regard, the paper provides an interesting application case, which is a positive feature of the research.

Author Response

(The authors gave the same response as above.)

Round 2

Reviewer 3 Report

The provided suggestions have been properly addressed and the manuscript has been strengthened. Congratulations to the authors for their efforts.